# Differential Effects of Human P301L Tau Expression in Young versus Aged Mice

**DOI:** 10.3390/ijms222111637

**Published:** 2021-10-28

**Authors:** Holly C. Hunsberger, Sharay E. Setti, Carolyn C. Rudy, Daniel S. Weitzner, Jeremiah C. Pfitzer, Kelli L. McDonald, Hao Hong, Subhrajit Bhattacharya, Vishnu Suppiramaniam, Miranda N. Reed

**Affiliations:** 1Division of Systems Neuroscience, Research Foundation for Mental Hygiene, Inc. (RFMH)/New York State Psychiatric Institute (NYSPI), New York, NY 10032, USA; hh2694@cumc.columbia.edu; 2Department of Psychiatry, Columbia University Irving Medical Center (CUIMC), NYSPI Kolb Research Annex, Room 736, 1051 Riverside Drive, Unit 87, New York, NY 10032, USA; 3Drug Discovery and Development, School of Pharmacy, Auburn University, Auburn, AL 36849, USA; ses0114@auburn.edu (S.E.S.); jcp0050@auburn.edu (J.C.P.); klm0114@auburn.edu (K.L.M.); szb0050@auburn.edu (S.B.); 4Center for Neuroscience Initiative, Auburn University, Auburn, AL 36849, USA; 5Department of Psychology, West Virginia University, Morgantown, WV 26506, USA; ccrudy@mix.wvu.edu (C.C.R.); dsweitzner@mix.wvu.edu (D.S.W.); 6Department of Pharmacology, Key Laboratory of Neuropsychiatric Diseases, China Pharmaceutical University, Nanjing 210009, China; honghao@cpu.edu.cn

**Keywords:** Alzheimer’s disease, glutamate, P301L tau, aging

## Abstract

The greatest risk factor for developing Alzheimer’s disease (AD) is increasing age. Understanding the changes that occur in aging that make an aged brain more susceptible to developing AD could result in novel therapeutic targets. In order to better understand these changes, the current study utilized mice harboring a regulatable mutant P301L human *tau* transgene (rTg(TauP301L)4510), in which P301L tau expression can be turned off or on by the addition or removal of doxycycline in the drinking water. This regulatable expression allowed for assessment of aging independent of prolonged mutant tau expression. Our results suggest that P301L expression in aged mice enhances memory deficits in the Morris water maze task. These behavioral changes may be due to enhanced late-stage tau pathology, as evidenced by immunoblotting and exacerbated hippocampal dysregulation of glutamate release and uptake measured by the microelectrode array technique. We additionally observed changes in proteins important for the regulation of glutamate and tau phosphorylation that may mediate these age-related changes. Thus, age and P301L tau interact to exacerbate tau-induced detrimental alterations in aged animals.

## 1. Introduction

Tauopathies are a group of neurodegenerative disorders involving the tau protein. Alzheimer’s disease (AD), the most common tauopathy, affects 5.4 million Americans and accounts for 60 to 80 percent of all dementia cases [1]. AD is characterized behaviorally by pervasive memory loss and cognitive deficits [2] and biologically by aggregates of beta-amyloid (Aβ) peptide and tau proteins, as well as neuron loss [3]. Though the proteins involved are known, the cause of AD is not, and there is no cure. While an overwhelming majority of animal research has focused on aberrant processing of Aβ as the initiating event for AD pathology, and for good reason [4,5], clinical drug trials targeting Aβ have been largely unsuccessful [6,7,8]. This obstacle necessitates parallel lines of research, including those focused on abnormalities in the tau protein and the downstream alterations produced by tau.

Our recent work suggests tau pathology results in high concentrations of extracellular glutamate via an increase in glutamate release and a decrease in glutamate transporters, leading to decreased uptake [9,10,11]. High concentrations of extracellular glutamate can lead to cell death through excessive activation, a process referred to as excitotoxicity [12]. Excitotoxicity is linked to several neurodegenerative disorders, including AD [13], and can occur when uncontrolled glutamate release surpasses the capacity of excitatory amino acid transporters (EAAT) to clear glutamate from the synapse. Because EAAT clearance of glutamate is the only way to terminate glutamate signaling, disturbances in this process can quickly lead to excitotoxic conditions [14]. Alterations in extracellular glutamate may explain the hyperexcitability of neural networks observed in those at risk for, or in the early stages of, AD and frontotemporal dementia, another tauopathy [15,16,17,18,19,20,21,22]. The role of tau in mediating neuronal hyperexcitability is supported by several animal studies showing that reductions in tau substantially reduce hyperexcitability in AD mouse models, induced-seizure models, and models of epilepsy [23,24,25,26,27]. Furthermore, our own work suggests reducing extracellular glutamate, via a decrease in glutamate release and an increase in uptake, results in improved learning and memory, decreased tau pathology, and increased PSD-95 expression [11]. Identification of factors that increase extracellular glutamate may have implications for the prevention and treatment of AD.

Although the cause of AD is unknown, there are several known risk factors for AD, the greatest of which is age. One way in which aging may potentiate the risk for AD is through an age-related increase in extracellular glutamate. There is an age-related decline in glutamate transporters and uptake [28,29], leading to higher levels of extracellular glutamate [30], and age-related hyperexcitability has been observed in rodents and humans [31,32,33]. Moreover, evidence suggests age-related increases in extracellular glutamate result in increased activation of extrasynaptic N-methyl-D-aspartate receptors (NMDARs) [8,34,35], which mediates tau phosphorylation [36] and tau-mediated cytotoxicity [37,38,39]. 

Researchers have attempted to reproduce the effects of aging in mice through the expression of transgenes containing mutations in *tau* known to promote pathological changes. While this approach replicates many aspects of AD pathology, the models are incomplete. An aged brain may enhance pathological changes in tau, and equally troubling, pathological changes in tau may exert different effects on an aged brain compared to a younger brain. Determining whether an aged brain differentially affects tau or whether tau differentially affects an aged brain is essential to increasing the validity of preclinical mouse studies. While preclinical animal models have been and will continue to be essential in developing new clinical therapies, most have been unsuccessful in predicting clinical efficacy. Successful translation requires an appropriate choice of animal model, including consideration of age. Using young and healthy animals to model human diseases that manifest in older age decreases their external validity. 

To date, studying the interaction between tau and an aged brain has been challenging because of the difficulty in separating the effects of prolonged mutant tau expression from those of an aged brain per se. Fortunately, mice harboring a regulatable mutant P301L human *tau* transgene (rTg4510; herein referred to as TauP301L), in which P301L tau expression can be turned off or on by the addition or removal of doxycycline (DOX) in the drinking water, can be used to assess the efforts of mutant tau expression during aging separate from prolonged expression. Because of the P301L mutation in tau, TauP301L mice exhibit a distinct, age-related profile of behavioral and pathological abnormalities observed in AD [40,41,42]. Because we can wait to turn on tau expression until the mice have aged, these mice allow us to study the effects of aging independently of prolonged mutant tau expression. Here, we compare the effects of 5 months of P301L tau expression on memory, glutamate regulation, synaptic proteins, and tau pathology using young, but developmentally mature, mice versus aged mice to test the hypothesis that aging will exacerbate deficits associated with P301L tau expression.

## 2. Materials and Methods

### 2.1. Mice

Creation of TauP301L mice has been described previously [42,43]. Briefly, mice expressing regulatable human four-repeat tau lacking the N-terminal sequences (4R0N) with the P301L mutation (herein referred to as ‘TauP301L’ mice) were created by crossing 129 s 6 mice expressing the tet-off tetracycline transactivator (tTA) reading frame placed downstream of Ca^2+^/calmodulin kinase II (CaMKII) promoter elements with FVB/N mice heterozygous for the tetracycline response element (TRE) TauP301L transgene [10,11,44]. Original breeding pairs were a generous gift from Dr. Karen Ashe and were bred and maintained at Auburn University. Groups consisted of young versus old TauP301L mice as well as their age-matched transgene negative littermates (herein referred to as ‘TgNeg’ mice), which were negative for both human *tau* and the promoter/activator element. Experimental male mice were group housed, between two and five per cage, in a temperature- and humidity-controlled colony room with a 12:12 light/dark cycle. All experimental procedures were conducted in accordance with the standards of International Animal Care and Use Committee, and the Auburn University Animal Care and Use Committee approved all experimental procedures used in the study.

### 2.2. Experimental Design

In the tet-off system, transcription and protein expression occurs when the tTA protein binds to TRE [45]. In the presence of doxycycline, a tetracycline derivative, the tTA protein cannot bind to TRE, allowing tau expression to be suppressed. The regulatable expression of the tau protein allowed us to compare P301L tau expression in young versus aged mice. To suppress tau expression [44], 40 ppm doxycycline hyclate (~6.66 mg/kg/day or 0.015 mmol/g/day of DOX) was administered via water bottles to breeder dams for three weeks prior to mating and to all experimental mice, including the age-matched TgNeg littermates that served as controls, from birth until 2.5 months of age for the “young” group or until 15 months of age for the “aged” group. At either 2.5 or 15 months of age, DOX was removed from the drinking water of all mice, and for tauP301L mice, tau expression began and continued for 5 months until 7.5 months of age in the young P301L mice and 20 months of age in the aged P301L mice. Thus, tau was expressed from 2.5 to 7.5 months of age in the “young” P301L group versus 15–20 months of age in the “aged” P301L mice, resulting in four groups: Young-TgNegs (*n* = 12), Young-P301Ls (*n* = 15), Aged-TgNegs (*n* = 25), and Aged-P301Ls (*n* = 18). 

Cognitive testing included a y-maze spontaneous alternation test and Morris water maze (MWM) task. Mice underwent visible platform training approximately 30 min after the last MWM probe trial to assess visual and motoric functioning. After visible platform training, mice underwent in vivo anesthetized glutamate recordings and were euthanized immediately afterwards.

### 2.3. Morris Water Maze (MWM)

The MWM was performed as previously described [11] in a 12″H × 47″D pool. For hidden platform training, the mouse was released from a semi-random starting location and was allowed 60 s to locate the platform, where the mouse remained for 15 s. The next trial was initiated after an inter-trial interval of 20–25 min to allow for adequate recovery of aged mice and to reduce the likelihood of confounds, such as deficits in swim speed. A probe trial was conducted 24 h after the last hidden platform training trial. For hidden training trials, we compared the latency to find the hidden platform and swim speed, as well as pathlength to the hidden platform, a measure generally considered to be immune to differences in swimming speed [46]. For probe trials, the percent time in the target quadrant and platform-crossing index (PCI) was examined.

### 2.4. Visible Platform Test

The visible platform test was performed at the end of MWM testing to ensure any deficits observed in the MWM were not due to visual or motor deficits [11]. Each mouse started facing the wall and was given 2 min to swim to the platform where it remained for 20 s. After each trial, the platform was moved along the back of the tub wall. All mice were given 5 independent trials with approximately 10 min between each trial.

### 2.5. Y-Maze Spontaneous Alternation Test

The Y-maze was performed as previously described [47]. During testing, mice were placed in the start arm facing the wall and allowed to explore the maze for 8 min. An arm entry was defined as all four paws entering the arm. The total number of arm entries and the time to start exploring the maze was also examined to ensure equal exploration across the groups. One successful alternation was defined as entering each of the three arms consecutively (1, 2, 3 or 3, 2, 1, etc.). Alternation behavior was calculated as: (number of alternations/(total number of arm entries − 2)) × 100. The floor was cleaned with 1% acetic acid solution after each mouse.

### 2.6. Enzyme-Based Microelectrode Arrays (MEAs)

To examine glutamate dysregulation, microelectrode arrays (MEAs) with four platinum recording sites were used along with a reference electrode (Ag/AgCl) placed in the opposite hemisphere (Quanteon, L.L.C., Nicholasville, KY, USA) [48]. Coating of the microelectrodes has been described previously [10,11,49]. Each electrode was calibrated prior to use and a standard curve for the conversion of current to glutamate concentration was generated. A single barrel glass micropipette (Quanteon) was attached to the MEA assembly, 80–100 μm away from the MEA surface, to locally apply potassium chloride (KCl: 70 mM KCl, 79 mM NaCl, 2.5 mM CaCl_2_, pH 7.4) or glutamate (200 μM, pH 7.4). The Picospritzer III (Parker-Hannifin, Cleveland, OH, USA), attached to the micropipette, was set to deliver volumes of 50–250 nL.

### 2.7. In Vivo Anesthetized Recordings

Recordings were performed as previously described [10,11,49]. Mice were anesthetized with isoflurane (1–4% continuous inhalation), placed in a stereotaxic device (David Kopf Instruments, Tujunga, CA, USA), and recordings were taken in all three subregions of the hippocampus (DG (AP: −2.3 mm, ML: ±1.5 mm, DV: 2.1 mm), CA3 (AP: −2.3 mm, ML: ±2.7 mm, DV: 2.25 mm), CA1 (AP: −2.3 mm, ML: ±1.7 mm, DV: 1.4 mm)) [50]. All MEA recordings were performed at 10Hz. Tonic levels of glutamate were measured and averaged over 10 s prior to KCl or glutamate injections, which were performed only after a stable baseline was reached. Evoked release of glutamate was measured by applying KCl every 2–3 min, as previously described [51,52,53], and was used to measure the “capacity” of the nerve terminals to release glutamate [54]. The evoked signal is reproducible, which indicates a functioning neuronal glutamate system [55]. Average amplitude was determined by subtracting the peak amplitude from the basal measure prior to stimulus ejection. After 10 reproducible signals were evoked, the MEA was repositioned into a new hippocampal subfield until all 3 subregions (DG, CA3, and CA1) were examined. In the opposite hemisphere, exogenous glutamate was applied to examine glutamate uptake or clearance in all 3 subregions of the hippocampus. Varying volumes (50–250 nL) of 200 μM sterile-filtered glutamate solution were applied every 2–3 min. Because glutamate transporters exhibit Michaelis–Menten kinetics [56], a range of volumes (50–250 nL) of exogenous glutamate was injected to expose differences in uptake as previously described [49]. Both hemispheres used for drug injection, and sub-regions within a hemisphere, were counterbalanced. The glutamate signals in each hippocampal subfield were averaged into a representative signal for comparison and analyzed using a custom Microsoft excel software program (MATLAB; Natick, MA, USA).

### 2.8. Immunoblotting

Immunoblotting has been described previously [10,11]. A bicinchoninic acid (BCA; Thermo Fisher Scientific, Waltham, MA, USA) protein assay using bovine serum albumin (BSA; Sigma, St. Louis, MO, USA) as a standard was used to determine protein concentrations. Hippocampal protein samples were heated for 5 min to 95 °C (GLT-1, actin, CP13, MC1, and PHF1) or not heated (vGLUT1, PSD95, Tau-5, HT7, synaptophysin, actin, GSK3β, pGSK3β) and loaded on 10% hand-cast sodium dodecyl sulfate (SDS)-page gels. After transfer, membranes were blocked for 1 h at room temperature (∼23 °C) and then incubated with a primary antibody (see Table 1 for list of antibodies) directed against the protein of interest overnight at 4 °C. The next day, membranes were incubated with the appropriate biotinylated or horseradish peroxidase conjugated secondary antibody for 1 h at room temperature (∼23 °C). Blots were then incubated with SuperSignal West Pico chemiluminescent substrate (Thermo Scientific, Waltham, MA, USA) or Novex AP chemiluminescent substrate (Invitrogen) for 5 min and visualized using Fluorchem E imager (Cell Biosciences, Preston VIC, Australia).

### 2.9. Data Analysis

All statistical analyses were performed using JMP (SAS, Cary, NC, USA). Statistical analysis consisted of ANOVA and repeated-measures ANOVA (RMANOVA). For all measures, the main effects of transgene (Tg; TgNegs vs. TauP301L) and Age (Young vs. Aged), as well as the interaction between the two (Tg*Age) were assessed. For the two-way RMANOVA of behavioral data, Trial or Probe served as the within-subject variables. All significant omnibus tests were followed by Fisher’s LSD post hoc comparisons. The critical alpha level was set to 0.05. All values in the text and figures (Prism GraphPad, San Diego, CA, USA) represent means ± SEM.

## 3. Results

### 3.1. Tau Expression Is Similar in Young vs. Aged Mice

One caveat of the tet-off model is that DOX efficacy is closer to 85–90% [41,42,44], and thus, a slow “leakage” of tau expression can occur, even in the presence of DOX. Though we have previously shown this “leakage” to be insufficient to produce memory deficits in young P301L mice [11], it is possible this low-level tau expression may eventually result in increased total tau in older mice on DOX for a longer period of time. In addition, chronic suppression of the tTA system with DOX has been shown to result in less protein expression when DOX is withdrawn, though this effect is observed when DOX is delivered at 200 ppm as compared to the 40 ppm used here [42]. Nevertheless, to ensure tau levels were similar in the young and aged P301L groups, we compared human tau levels using HT7 before proceeding with other analyses and observed no difference between the P301L groups (*p* > 0.05; Figure 1).

### 3.2. Age and P301L Tau Expression Alter Learning and Memory in the Morris Water Maze and Y-Maze

#### 3.2.1. Visible Platform Training

To ensure that mice were similar in visual and motor abilities, visible platform training was conducted at the conclusion of MWM testing. Three Aged-TgNegs were identified as having visible platform times greater than two standard deviations above those of Young-TgNegs, and thus, were removed from analysis of memory tasks [57]. After their removal, there were no differences in latency to find the platform among the groups (*ps* > 0.05).

#### 3.2.2. Morris Water Maze

To assess age- and transgene-related alterations in spatial learning and reference memory, young and aged mice were examined using the Morris Water Maze. During hidden platform training, latency to reach the hidden platform during MWM acquisition did not differ among the groups on Day 1 (*p*s > 0.05). On Days 2 and 3, however, latency to find the platform was longer in Aged-P301L mice compared to all other groups (Figure 2A; Tg: F (1,60) = 18.9, *p* < 0.0001 and Age: F (1,60) = 11.16, *p* = 0.0014). This difference was not due to differences in swim speed, as Aged-P301Ls did not differ in swim speed (Figure 2B, *p*s > 0.05). Nevertheless, we also examined pathlength, a measure generally considered to be immune to differences in swimming speed [46]. P301L mice, regardless of age, exhibited significantly longer pathlengths to locate the hidden platform during MWM acquisition (Figure 2C; Tg: F (1,60) = 14.16 *p* = 0.0004). In particular, the pathlength of Aged-P301Ls on Days 2 and 3 was significantly longer than that of both the Young-TgNegs and Aged-TgNegs (*p*s < 0.05) but not the Young-P301Ls (*p* > 0.05). Additionally, pathlength on Days 2 and 3 of hidden platform testing were longer in aged mice, regardless of Tg status (Figure 2C; Day × Age: F (2,59) = 4.09, *p* = 0.021). To further assess the differential acquisition among young and aged mice, a difference score between Day 1 and Day 3 pathlength was computed. Young mice, regardless of Tg status, had larger difference scores than aged mice (Age: F (2,61) = 3.51, *p* = 0.019; Young: 599 ± 59; Aged 426 ± 49), indicating faster acquisition. Importantly, there were no differences in pathlength among the groups on Day 1 (*p*s > 0.05), suggesting differences in acquisition on Days 2 and 3 were likely due to differences in learning and not motoric or visual functioning.

During the probe trial, impaired performance for both TauP301L and aged mice was evident in the time spent in the target quadrant (Figure 2D; Tg: F (3,60) = 5.74, *p* = 0.034; Age: F (3,60) = 5.74, *p* = 0.0004). Notably, Aged-TgNegs exhibited significantly less time in the target quadrant relative to Young-TgNegs (*p* < 0.05), and Aged-P301Ls spent less time in the target quadrant relative to both Young-TgNegs and Young-P301Ls (*p*s < 0.05). For platform crossing index (PCI), Aged-P301Ls exhibited significantly impaired performance compared to all other groups (Figure 2E; *ps* < 0.05), and their performance was not significantly different from chance, defined as 25% for percent time in target quadrant (*p* = 0.81) and zero for PCI (*p* = 0.10). In contrast, the other three groups performed significantly better than chance for both percent time in target quadrant and PCI (*ps* < 0.05). Together, these data indicate that P301L expression during aging exacerbates deficits in spatial reference memory to P301L expression when mice are younger.

#### 3.2.3. Y-Maze Spontaneous Alternation Test

We next used a free-running or continuous spontaneous alteration y-maze to measure spatial working memory as previously described [58]. This task is particularly sensitive task to age-related deficits [59,60,61]. We first ensured there were no differences in locomotor activity or motivation to explore the maze by measuring the total number of arm entries or latency to begin exploring the maze, respectively [62]. There were no differences among the groups for either measure (*p*s > 0.05; Figure 3A,B), suggesting similar levels of locomotor activity and motivation to explore. We next examined spatial working memory by measuring the number of alterations made during the 8 min trial, where alternation is defined as the successive entries of the three arms on overlapping triplet sets, in which three different arms are entered [58]. Alternations were reduced in aged mice (Figure 3C; Age: F (3,64) = 2.71, *p* = 0.015), particularly for Aged-P301Ls compared to Young-TgNegs (*p* < 0.05), again suggesting that aged brain exacerbates deficits associated with P301L expression.

### 3.3. Age- and Tau-Associated Changes in Glutamate Regulation

Previous work suggests age-related changes in hippocampal glutamate regulation [63], and we have previously shown that glutamate dysregulation mediates the memory impairment observed in young mice expressing mutant P301L tau [10,11]. Here, we sought to determine whether P301L tau expression in aged mice might differentially affect glutamate regulation when compared to P301L tau expression in young mice. As previously reported [11], P301L expression in young mice increased tonic glutamate levels in the CA3 and CA1 subregions compared to Young-TgNegs (*p*s < 0.05), and no differences were observed between the Young-P301Ls and Young-TgNegs in the DG subregion (Figure 4A–C; *p* > 0.05). Rather, for the DG subregion, aged mice, regardless of Tg status, exhibited higher tonic glutamate levels (Figure 4A; Age: F (3,37) = 4.05, *p* = 0.001). For the CA3 subregion, the pattern in aged mice was opposite that of their young counterparts, such that Aged-TgNegs exhibited higher tonic glutamate than Aged-P301Ls (Figure 4B; Tg × Age: F (3,38) = 1.88, *p* = 0.040). In the CA1, P301L mice, regardless of age, exhibited higher tonic glutamate (Figure 4C; Tg: F (3,39) = 3.57, *p* = 0.001), though there was a trend towards higher levels in Aged-TgNegs relative to Young-TgNegs (*p =* 0.07).

To examine the capacity for glutamate release [54], KCl was injected via a micropipette attached to the MEA. For evoked release in the DG, a pattern similar to that of tonic glutamate was observed; aged mice, regardless of Tg status, exhibited greater evoked release (Figure 4D; Age: F (3,33) = 3.59, *p* = 0.013). In contrast, in the CA3 (Tg: F (3,34) = 4.06, *p* = 0.015; Age: F (3,34) = 4.06, *p* = 0.041) and CA1 (Tg × Age: F (3,35) = 3.61, *p* = 0.043), evoked release was greater in the P301L and aged groups compared to Young-TgNegs (Figure 4E,F).

To examine glutamate clearance, exogenous glutamate was applied, and uptake was monitored by measuring the net area under the curve (AUC). For all three subregions, net AUC was higher, indicating a reduction in uptake, in Young-P301Ls compared to Young-TgNegs (*p*s < 0.05), as previously described [10,11]. In the DG, however, this pattern was reversed in aged mice (Figure 4G; Tg × Age: F (3,32) = 2.76, *p* = 0.014). In the CA3 (Figure 4H; Tg: F (3,33) = 2.90, *p* = 0.036), the pattern of aged mice was similar to that observed in young mice such that Aged-P301Ls exhibited a greater net AUC than Aged-TgNegs (*p* < 0.05). For the CA1 (Figure 4I; Age: F (3,35) = 2.89, *p* = 0.026), net AUC was greater in both aged groups compared to Young-TgNegs (*p*s < 0.05). Together, these results suggest both age and P301L tau expression alter glutamate regulation and reveal specific subregional differences associated with these changes.

### 3.4. Age- and Tau-Associated Changes in Protein Expression

#### 3.4.1. Tripartite Synapse

Because vesicular glutamate transporter (vGLUT) expression has been shown to mediate glutamate release [64], we sought to determine whether increases in vGLUT1 expression might explain the age- and tau-related perturbations in glutamate release. As previously reported [10,11], P301L tau expression in young mice increased vGLUT1 expression (*p* < *0*.05). However, while Aged-TgNegs also exhibited an increase in vGLUT1 levels, Aged-P301Ls did not (Figure 5A; Tg × Age: F (1,29) = 70.6, *p* < 0.0001). Our previous work suggests the increase in vGLUT1 expression in Young-P301Ls is not due to a widespread increase in pre-synaptic terminals [10,11]. To determine if the same was true for Aged-TgNegs and whether the lack of increase in Aged-P301Ls was due to a loss of pre-synaptic terminals, synaptophysin immunoblotting was performed. There were no differences in synaptophysin among the groups with the exception of the Aged-P301Ls (Figure 5B; Tg × Age: F (1,28) = 250.4, *p* < 0.0001), for whom synaptophysin expression was significantly decreased. This decrease in synaptophysin is indicative of an overall decrease in pre-synaptic terminals that may account for the decrease in vGLUT1 expression in Aged-P301Ls compared to Young-P301Ls. Synaptophysin expression in Young-P301Ls and Aged-TgNegs did not differ from the Young-TgNegs (*p*s > 0.05), suggesting the increase in vGLUT1 expression in these two groups was not due to an increase in pre-synaptic terminals. Finally, the ratio of vGLUT1 expression to synaptophysin expression was examined to determine if the relative amount of vGLUT1 in pre-synaptic terminals differed among the groups. Relative to the Young-TgNegs, all groups displayed a significantly greater vGLUT1 to synaptophysin ratio (Figure 5C; Tg × Age: F (1,29) = 10.82, *p* = 0.003), indicating greater vGLUT1 levels in pre-synaptic terminals.

We next examined the effects of tau and age on PSD-95, a major postsynaptic scaffold protein at glutamatergic synapses often used as a marker of excitatory synapses. Both Young-P301Ls and Aged-P301Ls exhibited a greater than 50% reduction in PSD-95 expression relative to their age-matched TgNeg littermates (Figure 5D; Tg: F (1,29) = 397.1, *p* < 0.0001), suggesting a possible loss of excitatory synapses. There was no change in PSD-95 expression for Aged-TgNegs compared to Young-TgNegs (*p* > 0.05), and no differences among the groups for the loading control, beta-actin (*p*s > 0.05).

In many neurodegenerative diseases including AD, the expression of astrocytic glutamate transporters, which remove glutamate from the extracellular space, is decreased [65,66]. To determine whether the impaired glutamate uptake we observed in aged and P301L mice is associated with a loss of glutamate transporters, we examined hippocampal GLT-1 expression, the major transporter responsible for removing glutamate from the extracellular space in rodents [67]. Relative to Young-TgNegs, all three groups exhibited a greater than 50% reduction in GLT-1 expression (Figure 5E; Tg × Age: F (1,26) = 98.7, *p* < 0.0001).

#### 3.4.2. Tau Pathology

To determine if an aged brain exacerbates tau pathology, we used three antibodies (MC-1, CP-13, and PHF1) that recognize phosphorylation- and conformation-specific changes in tau. For all three antibodies, there was a main effect of transgene (MC-1: F (1,25) = 145.4, *p* < 0.0001; CP-13: Tg: F (1,28) = 53.3, *p* < 0.0001; PHF-1: F (1,29) = 55.9, *p* < 0.0001) such that P301L mice, regardless of age, exhibited greater tau pathology (Figure 6). There were no differences between the Young-TgNegs and Aged-TgNegs for any antibody (*p*s > 0.4).

Because our primary goal was to discern whether aging exacerbated tau pathology, we next compared tau pathology in just the Young-P301Ls and Aged-P301Ls. For MC-1, which recognizes an early conformational change in tau, there was no difference between Young and Aged P301L mice (*p* = 0.51; Figure 6A). We also did not observe any differences in CP-13 expression (*p* = 0.2; Figure 6B), which recognizes one of the earliest phosphorylation changes in AD (pSer202). Interestingly, when we compared the groups using PHF-1, a late-stage marker of tau pathology at Ser396/404 [41], we observed increased expression in the Aged-P301Ls compared to the Young-P301Ls (*p* = 0.05; Figure 6C), suggesting an aged brain may exacerbate tau pathology.

#### 3.4.3. Glycogen Synthase Kinase-3-Beta

Previous work has demonstrated an age-related increase in the immunoreactivity of glycogen synthase kinase-3-beta (GSK3β) [68], a kinase that phosphorylates tau [69]. We have observed a similar increase in GSK3β activity in young P301L mice [11]. To determine whether aging exacerbates the P301L-mediated increase in GSK3β activity, we examined both total GSK3β levels and phosphorylation of GSK3β at Ser9 (pGSK3β), which inhibits GSK3β activity and is used as an indirect marker of GSK3β activity [69]. We observed an age-related increase in total GSK3β expression (Figure 7A,B; Age: (1,29) = 113.6, *p* < 0.0001). As we previously reported [11], total GSK3β levels were not increased in Young-P301Ls relative to Young-TgNegs (*p* > 0.05). We next assessed GSK3β activity by measuring pGSK3β. Though total GSK3β levels were not altered in Young-P301Ls, pGSK3β levels were significantly decreased relative to Young-TgNegs (*p* > 0.05), indicating greater GSK3β activity, similar to our previous report [11]. In addition, aged mice, regardless of Tg status, also exhibited decreased pGSK3β expression, indicating an age-related increase in GSK3β activity (Figure 3A,C; Tg × Age: F (1,21) = 7.8, *p* = 0.011). In agreement, the ratio of pGSK3β to GSK3β was decreased in the three groups relative to Young-TgNegs (Figure 3A,D; Tg × Age: F (1,19) = 10.7, *p* = 0.004).

## 4. Discussion

Because AD lacks a cure or any disease-modifying therapeutics, it is critical to understand alterations that may be permissive for the development of AD. Understanding these alterations may identify potential targets for the development of treatment of AD. It has been well established that one of the greatest risk factors for developing AD is increasing age [70]. The mechanisms by which aging is the greatest risk factor for the development of AD are still under investigation. One of the age-related alterations that may exacerbate the risk for developing AD are detrimental increases in neuronal activity, termed hyperexcitability, that are driven in part by dysregulation of glutamatergic neurotransmission [71,72,73]. Studies on rodent models have described age-related increases in hippocampal glutamate levels and glutamatergic release [74,75,76].

The literature also suggests that hyperexcitability may also play a role in the development of AD. Patients with recurrent seizures are at increased risk for developing AD [77,78] and advanced AD is associated with increased risk for seizures [79,80,81]. Furthermore, evidence in the literature demonstrates an interaction between aging and hyperexcitability. For example, in patients with amnestic MCI, which is sometimes considered to be a predecessor to AD [82], patients exhibit hippocampal hyperexcitability associated with performing episodic memory tasks [83,84]. Given these findings, we sought to better understand the interaction between aging, the presence of AD-related pathology such as pathological tau, and hyperexcitability.

Similar to our previous work [10,11,44], we observed that P301L tau expression resulted in the presence of markers of pathological tau, as well as corresponding glutamate dysregulation, and deficits in learning and memory. However, we also determined that P301L tau expressed during aging exacerbated late-stage pathology resulted in enhanced deficits in learning and memory, as well as a loss in presynaptic terminals. Thus, we observed not only an impact of P301L expression, but a unique impact when expressed during aging, suggesting that there is a difference between the way an aged brain responds to pathological tau versus a young brain. Understanding mechanisms that enhance the susceptibility of an aged brain to pathological tau could elucidate a new promising target for the treatment of AD-related pathologies. Unfortunately, because we only assessed male mice in this study, we are unable to determine whether there is an interaction between sex, aging, and glutamate dysregulation. While our previous work utilizing this model to assess P301L-induced cognitive deficits indicated no sex-related interactions in young mice [11,44], expression of tau during aging may have revealed sex disparities and will be examined in future studies. Additionally, in the model utilized in this study, rTg4510 only expresses mutant tau protein. Thus, we are unable to assess the potential that aging may mediate the interaction between beta-amyloid pathology and pathological tau. This model was also recently shown to have additional transgene insertion/deletion mutations that mediate at least part of the tauopathy-like phenotype observed in the rTg4510 mouse line that need to be considered [85]. It is worth noting that with prolonged doxycycline expression, there is a possibility that there will be tau “leakage” in tau-positive animals. However, previous work from our laboratory [11] and others [42] suggests that levels of “leaked tau” are not sufficient to cause either memory impairment or tau pathology. If anything, the prolonged use of doxycycline in the aged animals may have diminished the potential exacerbating effects of age, given recent studies indicating doxycycline may be a potential therapeutic agent for the treatment of AD [86].

We observed differences in glutamate regulation that varied by hippocampal sub-region. The neuroanatomy of the hippocampal subregions may lend a greater understanding to the reasons underlying these differences. The DG projects to the CA3 via the mossy fibers pathway. The CA3 projects unto itself via recurrent synapses (which make up about 95% of its connections), and also to the CA1 via the Schaffer collateral pathway. We observed an age-related increase in tonic glutamate levels and KCl-evoked release in the DG. Previous literature suggests that the DG may be susceptible to hyperexcitability due to an increase in responsiveness of individual synapses [87]. However, there was no age-related change in glutamate clearance in this region, indicating that aged mice may have preserved GLT-1 levels in the DG at the timepoint we examined. Therefore, an age-related increase in tonic levels may be due to increased vGLUT1 and a corresponding increase in glutamate release, which we did observe in Aged-TgNegs. Interestingly, while vGLUT1 levels are decreased in AD patients [88,89], similar to what we observed in aged-P301Ls, vGLUT1 is upregulated in aged and amnestic MCI patients [88,90], which may help to explain the observed age-related hippocampal hyperexcitability observed in patients [83,84].

The recurrent synapses of the CA3 make it particularly susceptible to excitotoxicity [91]. In regard to the CA3, we observed increased tonic glutamate in the young-P301Ls as well as Aged-TgNegs, but not in the Aged-P301Ls. Despite having comparable CA3 tonic glutamate to Young-TgNegs, Aged-P301Ls did exhibit enhanced KCL-evoked release in the CA3. In other AD rodent models, the literature suggests the potential for a decrease in neurons, but a greater responsiveness to stimuli in those that remain [87]. While we did not directly measure neuron loss, the protein expression data indicated a loss of presynaptic connections (i.e., synaptophysin levels) in aged-P301Ls, as well as a P301L-associated reduction in excitatory synapses (i.e., PSD95 levels), which could be due to overall reductions in the glutamatergic neurons themselves.

While MEA studies were region specific, the protein expression studies were performed using the whole hippocampus. Thus, this prevented a true correlation of MEA results with protein expression by region. However, we observed that the results in the CA1 most closely corresponded with our protein expression results. P301L expression resulted in increased tonic glutamate levels as well as reduced glutamate uptake. In agreement, we observed P301L- and age-related reductions in GLT-1. Though we did not observe enhanced vGLUT1 expression in aged-P301Ls, we did observe a greater vGLUT1 to synaptophysin ratio, suggesting more vGLUT1 for a given pre-synaptic terminal and supporting the hypothesis that tau-related increases in vGLUT1 meditate the enhanced KCL-evoked glutamate release observed. Thus, KCl-mediated glutamate release may be occurring via a different mechanism such as astrocytic release [92]. It is possible that changes in vGLUT2, which also mediates glutamate release [93], might explain the differences in glutamate release observed and should be examined in future studies.

In the current study, we did not observe any age-related changes in synaptophysin expression. Evidence in the literature has demonstrated similar results (i.e., no age-related changes in hippocampal synaptophysin) [94,95] as well as age-related decreases in synaptophysin [96,97]. Our PSD-95 protein expression results also indicated no age-related changes and likewise, the literature is similarly paradoxical in aging as well as AD status [98,99,100]. Like other alterations in hyperexcitability, there may be a varying spectrum of alterations observed at specific ages that may account for the discrepancy seen in the results. In any case, the age-induced changes observed here could indicate the beginning of alterations in hyperexcitability that may make an aged brain more susceptible to the development of tau pathology in the progression of AD.

The literature suggests that alterations in glutamatergic signaling that result in hyperexcitability can ultimately promote the spread of pathological tau [101]. Our MEA and protein results indicate a possibility of an age- and P301L expression-related increase in hyperexcitability; however, we cannot be certain as we did not directly assess hyperexcitability. Future work assessing hyperexcitability via electrophysiological measurements, for example, would allow us to directly determine the correlation between the changes we observed in the current study and resultant exacerbated neuronal activity. One of the limitations of this transgenic model is that it expresses P301L tau throughout the forebrain, precluding the ability to assess tau spread. A model that selectively expresses mutant tau in one of the first brain regions known to be affected in AD, such as the entorhinal cortex [102], would allow for a better understanding of the role of glutamate dysregulation in the spread of pathological tau.

We observed that all P301L mice exhibited deficits in pathlength to locate the hidden platform, regardless of age, indicating that P301L tau expression within an aged brain did not differentially impact acquisition. In this study, we only assessed the impact of P301L tau expression on cognitive performance after 5 months of expression. Our previous work indicated that 5 months of tau expression was significant to induce cognitive impairment in young-P301L mice. However, it is possible that we would have seen a greater separation in degree of performance deficit had we assessed the aged-P301L at a greater timepoint of tau expression. Moreover, the Morris water maze was designed to minimize confounds, such as motoric deficits, typically observed in aging. We have observed in our lab that using longer intertrial intervals, fewer and shorter trials, and a slightly smaller pool, as well as removing performance-incompetent mice [57,103], can mitigate confounds, such as longer swim speeds, related to aging. In any case, the aged-P301L mice did show exacerbated deficits in memory, as assessed during probe trials, compared to young-P301L mice. These results are in spite of the fact that P301L mice exhibited similar levels of tau expression regardless of age.

The differences in cognitive performance that were observed could be due to differences in severity of tau pathology. While aged-P301L mice exhibited similar levels of early stage tau pathology, they exhibited greater later stage tau pathology, as indicated by increased expression of PHF1. PHF1, which labels phosphorylation at epitopes Ser396 and Ser404, is thought to be a marker of presence of neurofibrillary tangles, and neurofibrillary tangles have been found to be correlated with the degree of cognitive deficits in several studies assessing humans [104,105,106]. Thus, the aged mice may have an overall greater degree of tau phosphorylation, which could underly their worsened memory performance. This increase in tau phosphorylation might be due to the age-related decrease in total GSK3β and pGSK3β levels observed in both the aged TgNeg and TauP301L mice.

## 5. Conclusions

To the best of our knowledge, this is the first study to assess the interaction of aging and pathological tau without the confound of prolonged mutant tau expression. Our results indicate that aging and P301L expression interact to result in glutamate dysregulation and exacerbated tau pathology. These alterations may underlie the enhanced cognitive deficits that we observed. Because we did not directly manipulate glutamate signaling, we are not able to directly conclude a cause-and-effect relationship between glutamate, tau pathology, and cognitive deficits in this study. Our previous work, however, indicates that reductions in glutamate signaling via treatment with riluzole attenuated cognitive impairment and tau pathology in the rTg4510 model [11]. The mechanisms by which this age-related exacerbation in tau pathology and glutamate regulation occurs warrant further investigation. Additionally, further studies would address the potential that age-related glutamate dysregulation could result in enhanced spread of pathological tau from brain region to brain region.

## Figures and Tables

**Figure 1 ijms-22-11637-f001:**
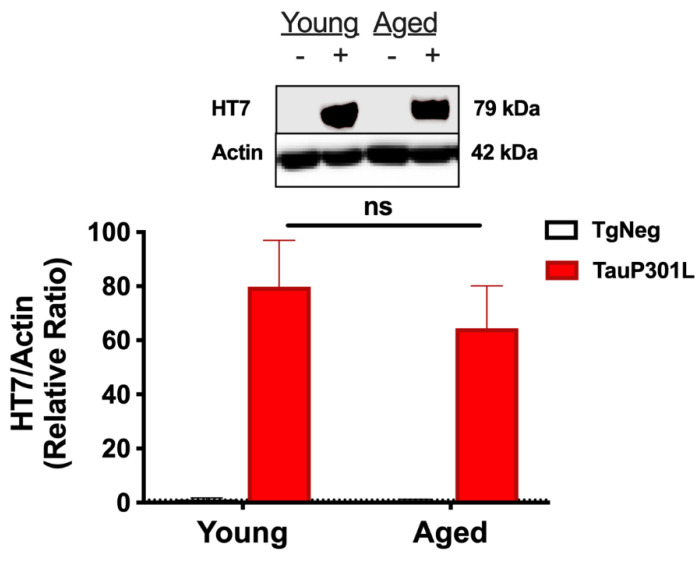
Tau expression. After DOX is removed, both Young- and Aged-P301L mice exhibit a strong tau expression in the hippocampus when examined using HT7, an antibody that recognizes only human tau. There is no difference between Young- and Aged- P301L tau expression for HT7. *n* = 5–7/group. − represents TgNeg and + represents P301L.

**Figure 2 ijms-22-11637-f002:**
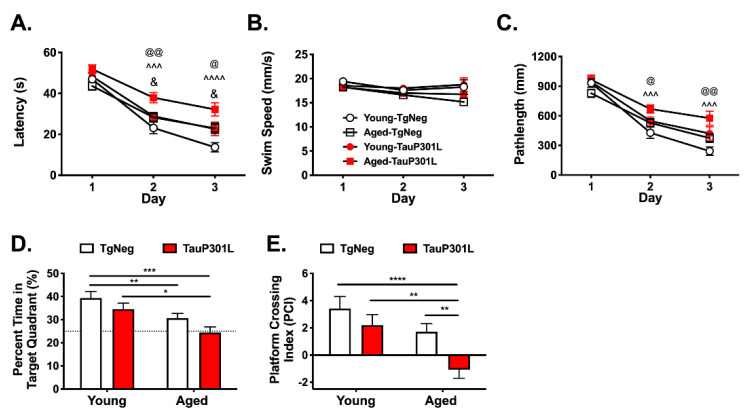
Aged-P301L mice exhibit deficits in the Morris water maze. Performance in hidden platform training was assessed by comparing (**A**) latency, (**B**) swim speed and (**C**) pathlength. Probe trial performance was assessed using percent time in the target quadrant (**D**) and platform crossing index (**E**). ^@^ represents significant difference between Aged-TgNegs and Aged-P301Ls; ^&^ represents significant difference between Young-P301Ls and Aged-P301Ls. *^/@/&^ *p* < 0.05, **^/@@^ *p* < 0.01, ***^/^^^^
*p* < 0.001, ****^/^^^^^ *p* < *0*.0001; *n* = 12–22/group.

**Figure 3 ijms-22-11637-f003:**
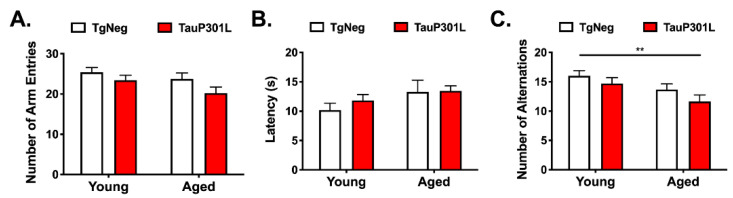
Aged-P301L mice exhibit deficits in the Y-Maze. No differences were observed among the groups for number of arm entries (**A**) or latency to begin exploring the maze (**B**), whereas the number of triads were reduced in Aged-P301L mice (**C**). ** *p* < 0.01; *n* = 12–22/group.

**Figure 4 ijms-22-11637-f004:**
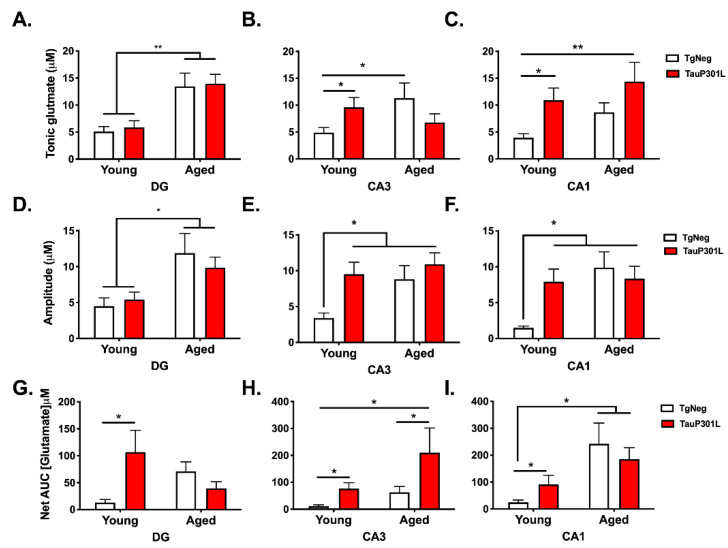
Age- and tau-associated changes in glutamate neurotransmission. Age and P301L tau expression resulted in alterations in measures of tonic, resting glutamate (**A**–**C**), KCl-evoked glutamate release (**D**–**F**), and glutamate uptake (**G**–**I**) in a subregion-dependent manner. * *p* < 0.05; ** *p* < 0.01; *n* = 8–13/group.

**Figure 5 ijms-22-11637-f005:**
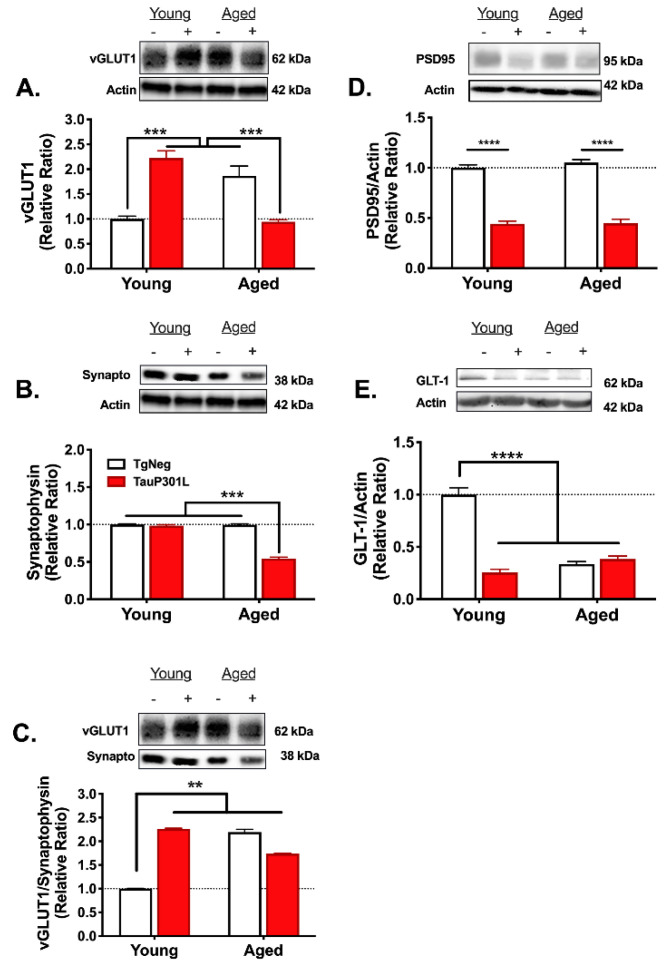
Age- and tau-associated alterations in the tripartite synapse. Age and P301L tau expression resulted in alterations in vGLUT1 (**A**), synaptophysin (**B**), vGLUT1/synaptophysin (**C**) PSD-95 (**D**), and GLT-1 expression (**E**). ** *p* < 0.01; *** *p* < *0*.001; **** *p* < *0*.0001; *n* = 7–9/group. − represents TgNeg and + represents P301L.

**Figure 6 ijms-22-11637-f006:**
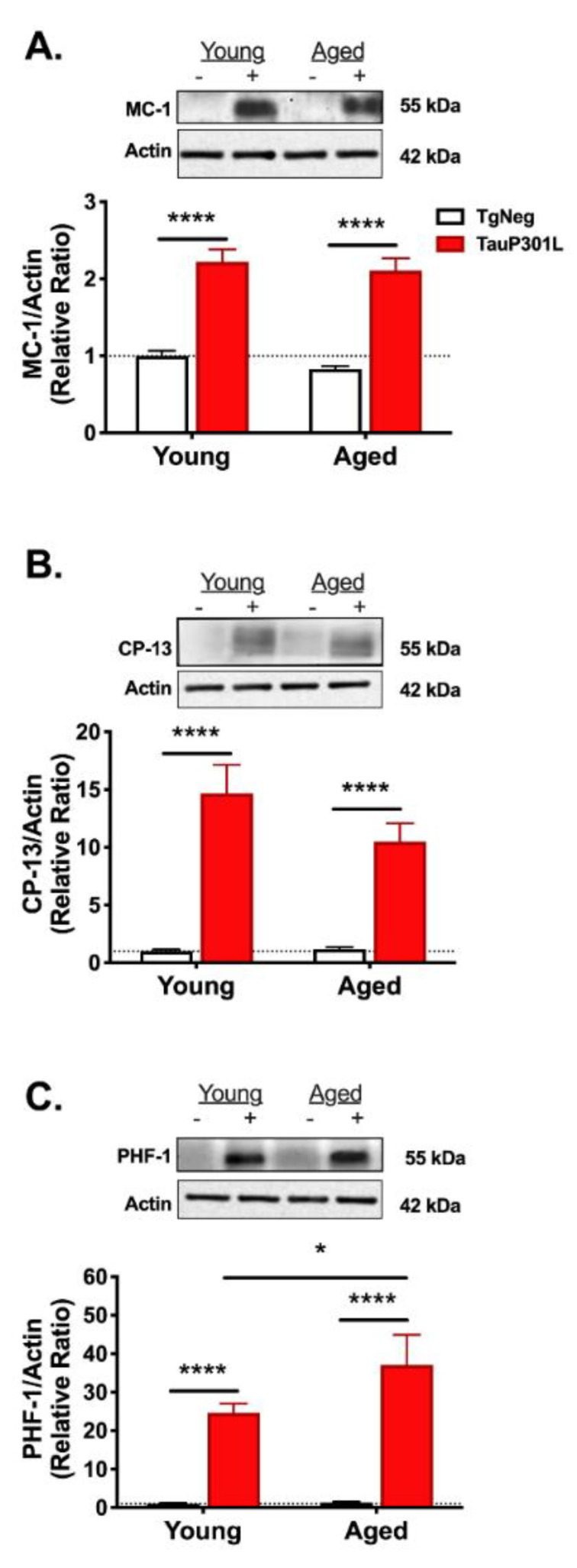
Tau pathology in young and aged mice. Levels of the conformation antibody MC-1 (**A**) and the phosphorylation antibody CP-13 (**B**) were similar between Young-P301L and Aged-P301L mice, whereas Aged-P301L exhibited increased staining with the late-stage antibody PHF-1 (**C**). * *p* < 0.05; **** *p* < 0.0001; *n* = 5–9/group. − represents TgNeg and + represents P301L.

**Figure 7 ijms-22-11637-f007:**
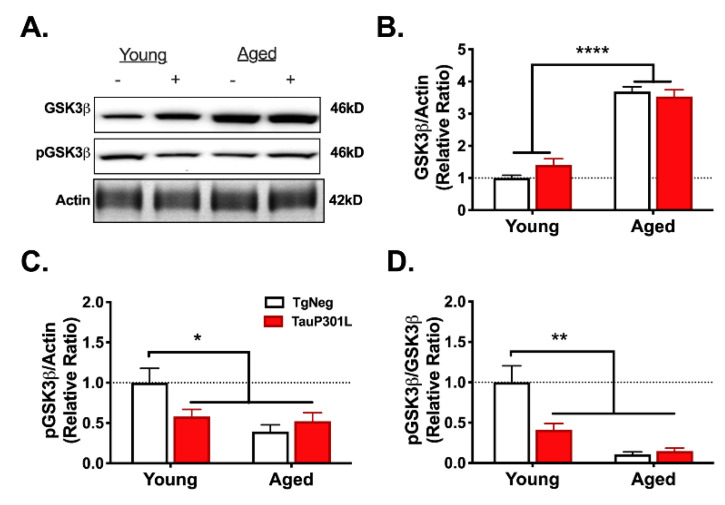
Age- and tau-associated changes in GSK3β activity. Age and P301L tau expression resulted in alterations in GSK3β (**A**,**B**), pGSK3β (**A**,**C**) and the ratio of pGSK3β/GSK3β (**A**,**D**). * *p* < *0*.015; ** *p* < *0*.01; **** *p* < *0*.0001; *n* = 5–9/group. − represents TgNeg and + represents P301L.

**Table 1 ijms-22-11637-t001:** List of antibodies.

Antibody	Amount Protein Loaded	Source
Tau-5	2 ug	Invitrogen
HT7	15 ug	Thermo Fisher
CP13	2 ug	Peter Davies
MC1	2 ug	Peter Davies
PHF1	2 ug	Peter Davies
vGlut1	2 ug	Millipore
Synaptophysin	2 ug	Sigma
GLT-1	20 ug	Millipore
PSD-95	20 ug	Millipore
GSK3β	20 ug	Cell Signalling
pGSK3β	20 ug	Cell Signalling
Actin	2–20 ug	Santa Cruz

## Data Availability

The datasets generated during and/or analysed during the current study are available from the corresponding author on reasonable request.

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
