# Peer review of "Differential Effects of Human P301L Tau Expression in Young versus Aged Mice"

_ijms, 2021, doi:10.3390/ijms222111637_

Round 1
Reviewer 1 Report
General comment:
The text is fluent and, generally, not very hard to follow, except, perhaps, in Results where it is often cluttered by passages referring to previous publications, justifying the present studies and/or interpreting the current findings. Such talk should be confined to Introduction or Discussion, making it easier to read and understand Results. See, for example section 3.3. on p. 8, but also elsewhere.
Specific queries and comments
# 1 Page 2, line 7 “excessive activation of glutamate receptors” Purists might ask - do you also mean mGluR’s? Perhaps just “excessive excitation” would be enough. It could also be interesting to acknowledge the researcher who coined the expression “excitotoxicity”. Was it John W Olney (doi: 10.1097/00005072-197503000-00005)?
# 2 Page 2, lines 7 and 8: My impression has been that it is not just “prolonged expression” but rather the presence of “pathological” or even “toxic” forms of tau which tend to somehow mess up the function of aging brains.
# 3 Page 4, paragraph 3, line 8; par. 4, l. 9 to 13. The concentration of K+ (70 mM) seems very high. In my experience such concentrations tend to result in gradually decreasing responses, possibly indicating irreversible changes in the release mechanism. Authors claim that the response was stable and cite Day et al. (2006) but there is no such publication in the list of references. Assuming that they meant the JNc paper (2006), we might perhaps choose to accept that the responses stay stable for two to five minutes but, in the present model, as I understand it, experimenters waited for 20 min (?) before they started to take their readings; did the release indeed stabilize and remain stable? Did researchers apply a series of stimuli similar to those shown in Day et al (2006) every 3 to 5 min? Perhaps the procedure could be explained more clearly, data shown more explicitly and the citation(s) checked and rectified (see also # 9). I may also add that, if gentler stimuli (say, closer to 30 mM K+) were used, authors perhaps would not have to worry about possible contribution from the release by astrocytes (see later in Discussion) which tends to occur at high levels of K+-stimuli.
# 4 Fig. 2B: My guess is, that the ages and sexes of animals were chosen so that they would correspond, in human equivalents, to, say, 30-year-old v. 80-year-old (or thereabouts). Isn’t it odd then that both groups of animals swam at the same pace?
# 5 Page 10, lines 3 to 4: What does vGlut2 do, then? I do understand that vGlut1 has been linked to glu release but the way it is mentioned here it sounds like a casual and possibly inaccurate statement disregarding vGlut2. It may be correct that vGlut1 is of crucial importance for some mechanisms of glu release in certain nerve endings but how would similar mechanisms function in vGlut2 containing nerve endings? Hippocampus may be a tricky place to study vGlut’s or use them as markers. Hippocampus contains both vGlut1 and 2 (depending on the origin of the fibres/nerve endings and vGlut1 and 2 might sometimes even co-localize in same cell structures. Both isoforms have their particular distributions (doi: 10.1111/jnc.15099; doi.org/10.1016/j.pneurobio. 2018.09.006) and can respond to changed experimental conditions (see e.g., doi.org/10.3390/biom11020294 for an example using the frontal cortex). Good practice would have been to look for changes in both vGlut1 and vGlut2. As it stands, the evidence is incomplete, and this fact should be acknowledged.
# 6 Fig. 5E: For a protein which is among the most highly expressed in brain, this is a very week signal, indeed. For which GLT1 variant was the antibody specific? This may be of crucial importance in a study related to Alzheimer disease (doi: 10.1385/NMM:3:2:105 and some of Rothstein’s papers, see also doi.org/10.1016/j.neuropharm.2019.03.002 for a review).
# 7 The study is based on testing and comparing various parameters in four groups of animals, resulting from a 2 x 2 matrix made up by combinations of Young and Aged v. PL301-expressing and TgNegs. Were all four groups given the doxycycline treatment? Or, was the Tg-negativity achieved by continuing administration of doxycycline (i.e Young/Doxy, Young/Non-doxy, Aged/Doxy, Aged/non-doxy where the non-doxy represents the groups where the use of doxycycline was discontinued and the offending human tau variant expressed as it seems to have been done in Fig 1?). Or did you compare doxy-naïve TgNeg (presuming that TgNeg means non-transgenic) v. “experimental” animals (i.e those doxy-treated PL301, from utero to 2.5 months or 15 months). An important control (TgNeg given doxy “treatment” v. TgNeg not given doxy “treatment”) could be missing here (more so, if the comparison was made between doxy-continuing and doxy-discontinued which, I hope, was not the case). This is not trivial in the light of recent publications suggesting that doxycycline could be used as a therapeutic agent in AD (doi: 10.3389/fphar.2019.00728). One could start by giving a clear description of the model (see also # 9) and stating explicitly which set of animals was taken as a control. The next step could be to introduce a doxycycline-free control, at least for the aged group. Alternatively, one should admit that doxycycline may have affected the validity of whole model, because all animals were, effectively, given a potential anti-AD therapy for large parts of their lives.
# 8 Fig. 1: With dox present, there is no tau expressed, it seems. Not even the normal rat tau which should be visible using Tau-5 ab. Isn’t it odd? Or is it just something confusing in the legend?
# 9 In 2.2. Experimental design, it cites Hölscher 1999 for the methodology of tau suppression by doxycycline. There is no mention of tau or doxycycline in that reference.
# 10 The dose of doxycycline is given in ppm. I would not know how to translate it to, say, millimole/g of body weight/day (so that we can better compare it with recommended therapeutic doses of doxycycline)?
Author Response
We thank the reviewer for the suggestions and have made all efforts to incorporate the suggested changes.
# 1 Page 2, line 7 “excessive activation of glutamate receptors” Purists might ask - do you also mean mGluR’s? Perhaps just “excessive excitation” would be enough. It could also be interesting to acknowledge the researcher who coined the expression “excitotoxicity”. Was it John W Olney (doi: 10.1097/00005072-197503000-00005)?
- We have removed ‘of glutamate receptors’ and added the suggested citation.
# 2 Page 2, lines 7 and 8: My impression has been that it is not just “prolonged expression” but rather the presence of “pathological” or even “toxic” forms of tau which tend to somehow mess up the function of aging brains.
- We have clarified this by saying ‘prolonged mutant tau expression’ throughout.
# 3 Page 4, paragraph 3, line 8; par. 4, l. 9 to 13. The concentration of K+ (70 mM) seems very high. In my experience such concentrations tend to result in gradually decreasing responses, possibly indicating irreversible changes in the release mechanism. Authors claim that the response was stable and cite Day et al. (2006) but there is no such publication in the list of references. Assuming that they meant the JNc paper (2006), we might perhaps choose to accept that the responses stay stable for two to five minutes but, in the present model, as I understand it, experimenters waited for 20 min (?) before they started to take their readings; did the release indeed stabilize and remain stable? Did researchers apply a series of stimuli similar to those shown in Day et al (2006) every 3 to 5 min? Perhaps the procedure could be explained more clearly, data shown more explicitly and the citation(s) checked and rectified (see also # 9). I may also add that, if gentler stimuli (say, closer to 30 mM K+) were used, authors perhaps would not have to worry about possible contribution from the release by astrocytes (see later in Discussion) which tends to occur at high levels of K+-stimuli.
- Demonstration of the reproducible signals at intervals as short as 15 s can be found in Day 2006 (Figure 5; doi/10.1111/j.1471-4159.2006.03673.x). We have added Day 2006 to the reference list and ensured all other citations are present. Citation issues occurred due to switching citation managers and having multiple users.
- 70 mM is the most commonly used concentration for MEA experiments (see also doi: 1038/s41598-020-71587-6; doi: 10.1093/gerona/glw088; https://www.ncbi.nlm.nih.gov/books/NBK2567/) and is used to examine the capacity for release, not necessarily physiological release. These additional citations have been added to the methods.
- We have previously used TBOA to rule out the role of astrocytic contribution, though not in the current study, which is why we acknowledge its potential role in the discussion. We agree that examining different K+ concentrations could be done in future studies.
- We have attempted to clarify that injections occurred only after a stable baseline was reached. KCl injections took place every 2-3 minutes, much longer than the interval at which reproducible signals can be observed in vivo (see Day 2006).
- We have attempted to clarify the methods and to add a citation to our previously published and very detailed method paper for using the MEA system - doi: 3791/55418
# 4 Fig. 2B: My guess is, that the ages and sexes of animals were chosen so that they would correspond, in human equivalents, to, say, 30-year-old v. 80-year-old (or thereabouts). Isn’t it odd then that both groups of animals swam at the same pace?
- Mice having visible platform times greater than two standard deviations above that of Young-TgNegs were removed from analysis of water-based tasks (see section 3.2.1.), which may have contributed to there being no difference in swim speed.
- Although we use pathlength as a primary dependent measure to avoid the potential confound of slower swim speeds, we have also tried to optimize our setup for aged animals. Notably, we use a smaller size pool than is sometimes used (12"H x 47" D), shorter trials (60 sec vs. 90-120 sec), and much longer intertrial intervals (20-25 min. vs. 60 sec). Together with our removal of performance-incompetent mice (see section 3.2.1 & doi: 1523/JNEUROSCI.22-05-01858.2002), we have observed that these procedural alterations reduce issues related to slower swim speeds in aged mice. It has been our experience that these procedures bias the results against our hypothesis, as including performance-incompetent mice and making the task for physically demanding tends to exacerbate the ‘age’ and ‘tau’ effects. Nevertheless, we feel reducing confounds, such as motoric deficits, to the best of our ability is critical to proper interpretation of memory tasks.
- Because we have not published these procedural optimization steps, we did not describe our anecdotal findings originally. Per request, we have now added a brief synopsis of these changes in the discussion section. We also added the pool dimensions to the methods and a note that the ITI was longer for the purpose of allowing adequate recovery in aged mice.
# 5 Page 10, lines 3 to 4: What does vGlut2 do, then? I do understand that vGlut1 has been linked to glu release but the way it is mentioned here it sounds like a casual and possibly inaccurate statement disregarding vGlut2. It may be correct that vGlut1 is of crucial importance for some mechanisms of glu release in certain nerve endings but how would similar mechanisms function in vGlut2 containing nerve endings? Hippocampus may be a tricky place to study vGlut’s or use them as markers. Hippocampus contains both vGlut1 and 2 (depending on the origin of the fibres/nerve endings and vGlut1 and 2 might sometimes even co-localize in same cell structures. Both isoforms have their particular distributions (doi: 10.1111/jnc.15099; doi.org/10.1016/j.pneurobio. 2018.09.006) and can respond to changed experimental conditions (see e.g., doi.org/10.3390/biom11020294 for an example using the frontal cortex). Good practice would have been to look for changes in both vGlut1 and vGlut2. As it stands, the evidence is incomplete, and this fact should be acknowledged.
- In the results, we changed the phrase “Because vesicular glutamate transporter-1 (vGLUT1) expression has been shown to mediate glutamate release” to be more general and to say “Because vesicular glutamate transporter (vGLUT) expression has been shown to mediate glutamate release”.
- In the discussion, we have noted that vlgut2 also mediates glutamate release (i.e., quantal size) and noted that vglut2 should be examined in future studies.
# 6 Fig. 5E: For a protein which is among the most highly expressed in brain, this is a very week signal, indeed. For which GLT1 variant was the antibody specific? This may be of crucial importance in a study related to Alzheimer disease (doi: 10.1385/NMM:3:2:105 and some of Rothstein’s papers, see also doi.org/10.1016/j.neuropharm.2019.03.002 for a review).
- We believe there are two reasons why our blots appear lighter than some published images. First, prior to running any blot, we run a dose (protein amount) -response curve to identify the linear range. We then select a protein and antibody concentration that ensures we can detect the hypothesized change (typically a decrease in protein expression), as well as the opposite effect in the case of a potential compensatory response. It has been our observation that many labs will load a protein amount (e.g., 40 ug) that does not allow for both increases and decreases to be detected. For example, an increase in protein expression may not be discernable because the difference between 20 and 40 ug of protein is not actually discernable (e.g., is outside the linear range). Thus, we tend to load protein amounts that are lower than those sometimes reportted.
- Second, we did not alter this image in any way, including even the acceptable ‘auto contrast’typical of most imaging systems/photoshop. While we do not feel doing so is wrong (and doing so generally ethically acceptable), we tend to avoid altering images as much as possible.
- Together, these methods result in lighter images. However, as both reviewers raised this concern, we have used alternative images or auto contrast to improve the quality of images shown for our western blots. Where alternative images were used, the stats were rerun as needed. No differences in interpretation occurred.
- Based on the website and a call to the company (Millipore), the GLT1 antibody (Uniport P43004, glutamate transporter 2) is not specific for a particular variant.
# 7 The study is based on testing and comparing various parameters in four groups of animals, resulting from a 2 x 2 matrix made up by combinations of Young and Aged v. PL301-expressing and TgNegs. Were all four groups given the doxycycline treatment? Or, was the Tg-negativity achieved by continuing administration of doxycycline (i.e Young/Doxy, Young/Non-doxy, Aged/Doxy, Aged/non-doxy where the non-doxy represents the groups where the use of doxycycline was discontinued and the offending human tau variant expressed as it seems to have been done in Fig 1?). Or did you compare doxy-naïve TgNeg (presuming that TgNeg means non-transgenic) v. “experimental” animals (i.e those doxy-treated PL301, from utero to 2.5 months or 15 months). An important control (TgNeg given doxy “treatment” v. TgNeg not given doxy “treatment”) could be missing here (more so, if the comparison was made between doxy-continuing and doxy-discontinued which, I hope, was not the case). This is not trivial in the light of recent publications suggesting that doxycycline could be used as a therapeutic agent in AD (doi: 10.3389/fphar.2019.00728). One could start by giving a clear description of the model (see also # 9) and stating explicitly which set of animals was taken as a control. The next step could be to introduce a doxycycline-free control, at least for the aged group. Alternatively, one should admit that doxycycline may have affected the validity of whole model, because all animals were, effectively, given a potential anti-AD therapy for large parts of their lives.
- While we noted in 2.2 that all experimental mice received treatment, we think this may not have been clear enough. Thus, we have explicitly stated that the age-matched TgNeg (now more clearly defined in section 2.1) littermates also received doxycycline. We believe this is clearer now, but please let us know if it is not.
- Unfortunately, we do not have any doxycycline-untreated groups for comparison. However, we have noted the possibility of the anti-AD therapy (which if anything made it less likely for us to see age-related exacerbations given the longer doxycycline exposure in the aged groups) to the discussion section, as suggested. The citation provided did not seem to be relevant to the work, though we believe we have cited the paper the reviewer meant to suggest. Please let us know if another citation is preferred.
# 8 Fig. 1: With dox present, there is no tau expressed, it seems. Not even the normal rat tau which should be visible using Tau-5 ab. Isn’t it odd? Or is it just something confusing in the legend?
- There is tau present with Tau-5 for all groups. However, because of the robust overexpression in P301L mice (13:1 human:mouse units), using blots where this can be readily observed means that the P301L groups have bands outside the linear range, a critique we have received in reviews of our prior work. As our primary concner is comparing the TauP301L groups, we used a short exposure and a protein concentration within the linear range. After careful consideration, we have decided to remove the Tau5 blot and use only the HT7 blot to prevent the concern raised here, as well as the issue of linear range raised previously.
# 9 In 2.2. Experimental design, it cites Hölscher 1999 for the methodology of tau suppression by doxycycline. There is no mention of tau or doxycycline in that reference.
- We appreciate the reviewer noting this. Our citations were altered when switching to a new citation manager. We have checked and edited the citations throughout.
# 10 The dose of doxycycline is given in ppm. I would not know how to translate it to, say, millimole/g of body weight/day (so that we can better compare it with recommended therapeutic doses of doxycycline)?
- Doxycycline = 444.4 g/mol
- 40 ppm = 90.01 uM = .09 mM
- Assuming a 5 ml water intake per 30g body weight per day, 40 ppm is estimated to result in 6.66 mg/kg/day (or .015 umol/g/day). These concentrations have been added to the methods.
Reviewer 2 Report
In my opinion the submitted manuscript is well-written and the subject is very interesting. I have only two minor suggestions:
- In the Introduction section importance of the presented studies should be emphasized more profoundly. Moreover, the Authors should briefly describe the tested parameters and study hypotheses.
- Number of the tested animals should be given in figure captions (Figure 2, Figure 3)
Author Response
We thank the reviewer for their suggestions and hope our efforts are well received.
In my opinion the submitted manuscript is well-written and the subject is very interesting. I have only two minor suggestions:
- In the Introduction section importance of the presented studies should be emphasized more profoundly. Moreover, the Authors should briefly describe the tested parameters and study hypotheses.
- The importance and hypotheses have been added.
- Number of the tested animals should be given in figure captions (Figure 2, Figure 3)
- These values have been added to the figure caption.
Reviewer 3 Report
This manuscript entitled “Differential Effects of Human P301L Tau Expression in Young Versus Aged Mice” has been reviewed sincerely. This work was well-performed and interesting data have been shown. Although I have no serious comments in this paper, there were several concerns as below.
- Several band density shown in WB data were very weak throughout paper. These results should be improved clearly.
- Mice age described in “Introduction” section should be moved in “M&M” section.
- GRURT1 and GSK3B1 expression in aged mice increased compared to that of young mice even in basal levels (Fig.5 and Fig. 7). Since these results are important, the mechanisms must be considered carefully.
Author Response
We appreciate the time taken to review our manuscript and hope that the changes meet the expectations.
This manuscript entitled “Differential Effects of Human P301L Tau Expression in Young Versus Aged Mice” has been reviewed sincerely. This work was well-performed and interesting data have been shown. Although I have no serious comments in this paper, there were several concerns as below.
- Several band density shown in WB data were very weak throughout paper. These results should be improved clearly.
- Improved blots have been used for Figure 5E, Figure 7A, and Figure 7B. please also see our response to reviewer #1 question #6.
- Mice age described in “Introduction” section should be moved in “M&M” section.
- The age information was added to section 2.2
- GRURT1 and GSK3B1 expression in aged mice increased compared to that of young mice even in basal levels (Fig.5 and Fig. 7). Since these results are important, the mechanisms must be considered carefully.
- We believe the reviewer means vGLUT1 in Fig. 5 when they say ‘GRURT1’, but please correct us if that is not the case.
- We have now added more information to the discussion section regarding the potential significance of increased VGLUT1 and GSK3beta in aged mice.
Round 2
Reviewer 1 Report
I am inclined to accept all explanations/clarifications given by authors as well as the changes in MS. One minor comment: Is O'Kane et al paper indeed the best reference to cite for the saturability of EAAT's? It focuses primarily on BBB glu transporters. The saturable nature of glutamate uptake in brain tissue itself has been known since the early 1970's ("high affinity" i.e. low Km); also in cultured glial cells. It is indeed surprisingly difficult to find studies which give values of kinetic constants of individual transporters in an explicit form - perhaps Mol Pharmacol 51 809-815 1997; it covers the two most important transporters (and their differences) in some detail. There could be even better references, possibly.
Reviewer 3 Report
None.